# The relationship between biodiversity and wetland cover varies across regions of the conterminous United States

Jeremy S. Dertien[ORCID]1*, Stella Self2, Beth E. Ross1,3, Kyle Barrett1, Robert F. Baldwin1

**1** Department of Forestry and Environmental Conservation, Clemson University, Clemson, South Carolina, United States of America, **2** Department of Epidemiology and Biostatistics, Arnold School of Public Health, University of South Carolina, Columbia, South Carolina, United States of America, **3** U.S. Geological Survey, South Carolina Cooperative Fish and Wildlife Research Unit, Clemson University, Clemson, South Carolina, United States of America

\* jdertie@clemson.edu

**Data Availability Statement:** All spatial data files are available from the Figshare database (DOI: 10.6084/m9.figshare.11968314).

## Abstract

Identifying the factors that determine the spatial distribution of biodiversity is a major focus of ecological research. These factors vary with scale from interspecific interactions to global climatic cycles. Wetlands are important biodiversity hotspots and contributors of ecosystem services, but the association between proportional wetland cover and species richness has shown mixed results. It is not well known as to what extent there is a relationship between proportional wetland cover and species richness, especially at the sub-continental scale. We used the National Wetlands Inventory (NWI) to model wetland cover for the conterminous United States and the National Land Cover Database to estimate wetland change between 2001 and 2011. We used a Bayesian spatial Poisson model to estimate a spatially varying coefficient surface describing the effect of proportional wetland cover on the distribution of amphibians, birds, mammals, and reptiles and the cumulative distribution of terrestrial endemic species. Species richness and wetland cover were significantly correlated, and this relationship varied both spatially and by taxonomic group. Rather than a continental-scale association, however, we found that this relationship changed more closely among ecoregions. The species richness of each of the five groups was positively associated with wetland cover in some or all of the Great Plains; additionally, a positive association was found for mammals in the Southeastern Plains and Piedmont of the eastern U.S. Model results indicated negative association especially in the Cold Deserts and Northern Lakes & Forests of Minnesota and Wisconsin, though these varied greatly between groups. Our results highlight the need for wetland conservation initiatives that focus efforts at the level II and III ecoregional scale rather than along political boundaries.

## Introduction

Evaluating the drivers of species diversity and distributions has been a central focus of ecology and is important for future conservation initiatives [1]. Numerous hypotheses have been

**Funding:** Margaret H. Lloyd-SmartState Endowment at Clemson University https://www.clemson.edu/cafls/lloyd_project/about/index.html The funders had no role in study design, data collection and analysis, decision to publish, or preparation of the manuscript.

**Competing interests:** The authors have declared that no competing interests exist.

posited to explain the spatial distribution of species and communities, none of which perfectly explain the hemispheric gradients of species richness [2]. Investigations of wildlife species richness drivers have found that the interaction of biotic and abiotic factors can determine species distributions and assemblages [3]. At the local ecosystem scale, the diversity of niche space, interspecific competition and anthropogenic land alterations can determine species richness and assemblage [4–7]. Increasing in extent to landscape and continental scales, factors such as temperature, annual precipitation, elevation, net primary productivity, and habitat heterogeneity act as drivers of species distributions [8–10]. Further, there is variation between different vertebrate classes as to the important environmental factors determining species richness distributions [2,3]. Increasing our understanding of the factors shaping large-scale biodiversity patterns would improve our ability to conserve and manage such diversity in the context of increasing anthropogenic global change.

Wetlands are globally recognized as important habitats for wildlife and human productivity [11–13]. Wetlands provide numerous ecosystem services including carbon sequestration, water filtration, nutrient retention and flood mitigation [14,15]. In addition, wetlands are important migratory stops for birds and mammals and breeding habitat for amphibians, birds, and some reptiles [16–19]. Studies at the local scale that compared across individual wetlands have found mixed results regarding the relationship between species richness and individual wetland area, with some studies finding a positive association between birds, mammals, and herptiles [20,21] and others finding no relationship between amphibians and wetland size [22,23]. Little is known on whether cumulative wetland area or proportional wetland cover (i.e., the portion of a pixel that is taken up by wetlands) at the landscape-scale, across an area as large as the conterminous United States, acts as a driver of sub-continental species richness patterns. Most amphibian species in the temperate zone have some reliance on aquatic systems during their life cycles [24], and as noted many bird species such as wading birds and waterfowl are obligate wetland occupants or rely on these habitats during migration [19,25]. A host of reptile and mammal species rely too upon wetlands for shelter, foraging, and breeding [14,26,27]; however, compared to amphibians and birds the relationship between wetland area and mammal and reptile species richness in the U.S. may be less clear as there is a greater proportion of high species richness coverage outside of regions with high probable wetland cover [28].

With the recognition for the importance of wetland habitats, there is increased attention to conserve wetlands for the joint maintenance of biodiversity and ecosystem services [29]. Some wetlands of the United States are protected by federal conservation laws and executive orders that regulate wetland depletion and incentives for the creation of some wetland habitat [30]. Substantial energy and research are also committed to the sampling and monitoring of the nation's wetland habitats [31]. However, these protections can be narrow in focus, especially as they do not offer protections to geographically isolated wetlands [32]. In addition, despite recent conservation efforts it is estimated that wetland area in the conterminous United States has decreased by 53% or approximately 47 million ha from the 1780s to the 1980s due to water diversion and land conversion [33].

Investigations of continental and global conservation importance, such as the relationship between proportional wetland cover and species diversity, are increasingly possible with greater computing capacity and advances in the availability of remotely sensed data. Advances in Bayesian spatial modeling and the associated computational procedures have enabled the estimation of spatially varying coefficients over a large spatial extent in a framework that allows for seamless inference. Such models allow for the creation of spatial layers that show how the relationship between a dependent and independent variable changes over space. These advancements are especially useful for complex socioecological systems, such as those that

surround wetland ecosystems where multiple biotic, abiotic and anthropogenic forces are influencing how wildlife species are spatially distributed. The ability to better understand how the influence of an environmental variable changes through space can allow for the spatially-explicit tailoring of conservation initiatives.

To better understand the relationship between wetland cover and large-scale species diversity patterns, we used a Bayesian spatial Poisson regression model to estimate the spatially varying relationship between wetland cover and the species richness of four different taxa of wildlife and for endemic species in the conterminous United States. We controlled for factors known to influence species diversity such as elevation, annual precipitation, and temperature. In addition, we estimated 10-year wetland change in the conterminous U.S. as a way of assessing how the change of proportional wetland cover could influence continental scale wildlife diversity. We predicted that birds, amphibians, and endemic species would have a positive association with proportional wetland cover across most of the U.S. given the previous findings of a positive relationship with wetland area [20,21]. This would be especially prominent in the southeast, due to higher niche availabilities and the higher species richness of these groups. We predicted that mammals and reptiles would also have a positive correlation with wetland cover; however, to a lower magnitude than birds and amphibians given the higher relative species richness of reptiles and mammals in the western U.S. and the likelihood of lower proportional wetland cover.

## Methods

### Spatial data processing

We acquired biodiversity data from BiodiversityMapping.org [28] which provides spatial data of cumulative species range maps across numerous wildlife and plant taxa including endemic wildlife species of the conterminous United States. Distribution maps of amphibians, birds, and mammals were updated in 2013 and was compiled using data from International Union for the Conservation of Nature [34] and BirdLife International [35]. Reptile data was generated in 2008 using data from NatureServe [36]. We downloaded U.S. wetlands vector data from the USFWS National Wetland Inventory Wetlands Mapper [37]. These data were separated by state or multiple shapefiles per state. We downloaded estimated average temperature data spanning 1980–1999 from the National Center for Atmospheric Research [38]. These data were a point shapefile in a grid with points spaced < 5 km apart across the conterminous United States. We acquired North American 1 km$^2$ elevation data from the U.S. Geological Survey [39] and 30-year average 1 km$^2$ precipitation data (1980–2010) from the PRISM climate group [40].

We added the endemic range maps of amphibians, birds, mammals, and reptiles into one cumulative endemics raster. We merged all state NWI wetland shapefiles into one feature class that encompassed all the wetlands of the conterminous United States, transformed the merged wetlands feature to a point layer, and deleted points that represented a wetland < 0.01 ha to ensure the removal of erroneous features. Finally, we removed all lake and river points as large lake/reservoir bodies positively biased estimates of wetland cover. In addition, other wetland types that terrestrial species are likely to use, such as freshwater emergent or freshwater forested/shrub wetlands, were often represented along or in close proximity to lake and river features.

We used a kernel density estimator with a search radius of 100 km to create a raster of proportional wetland cover with a cell size of 10 x 10 km to maintain the same spatial grain as the biodiversity data. Therefore, our wetland coverage was the estimated hectares of wetlands per that 100 km$^2$ cell. Given high spatial autocorrelation between average national temperature

data points, we used simple kriging on these data to calculate a continuous raster of average temperatures. The temperature, elevation and precipitation rasters were then transformed into simplified polygon features and spatially joined to the wetland feature to create one covariate spatial layer. In addition, we isolated the two-digit hydrologic unit codes (HUC2; n = 18) in the conterminous U.S. to incorporate contiguous biogeographic regions as a random effect [41]. We then analyzed the spatial correlation between all covariates using a Pearson's correlation cutoff of 0.7 between one covariate pair [42,43].

To calculate wetland change, we isolated woody and emergent herbaceous wetlands from the National Land Cover Database (NLCD) 2001 and NLCD 2011 and followed the same procedure as above to calculate proportional wetland cover. Then, we subtracted the modeled NLCD 2011 from the NLCD 2001 coverage to calculate 10-year wetland change. Finally, to calculate per cell percentage change of wetland cover we divided the NLCD 2001 wetland cover per cell by the estimate of 10-year wetland change. Data processing and analysis were conducted in ArcMap v. 10.5, ArcPro v. 2.1 (ESRI, Redlands, CA) and R (R Core Team 2019).

## Statistical analysis

Our statistical analysis was developed to simultaneously allow for spatially-varying patterns of responses to proportion of wetland cover while accounting for spatial autocorrelation. For each of the polygon features in our spatial covariate layer (n = 38,976), we let $y_s$ denote the species richness of the given taxonomic group at location $\boldsymbol{\ell}_s = (\ell_{s1}, \ell_{s2})$, the latitude-longitude location of the centroid of the $s$th polygon feature. We fit a Bayesian generalized mixed model to our polygon data [44,45]. We assume $y_s|\eta_s \sim Poisson(\lambda_s)$, that is, conditional on $\eta_s$, the observations independently follow a Poisson distribution with mean $\lambda_s = \exp(\eta_s)$. Here $\eta_s$ is a linear predictor defined by

$$\eta_s = \boldsymbol{z}_s \boldsymbol{\alpha} + x_s \beta(\boldsymbol{\ell}_s) + \gamma(\boldsymbol{\ell}_s).$$

Here $\boldsymbol{z}_s$ is a P-dimensional vector of covariates from location $\boldsymbol{\ell}_s$, including an intercept term, $\boldsymbol{\alpha}$ is the associated vector of fixed covariate effects, $x_s$ is the proportional wetland cover at location $\boldsymbol{\ell}_s$, $\beta(\boldsymbol{\ell}_s)$ is the effect of proportional wetland cover at location $\boldsymbol{\ell}_s$, and $\gamma(\boldsymbol{\ell}_s)$ is a spatial random effect at location $\boldsymbol{\ell}_s$. Note that the effect of wetland cover, the $\beta(\boldsymbol{\ell}_s)$ terms, depends on spatial location, allowing the effect of wetland cover on species diversity to vary from location to location. As wetland cover is known to impact species dynamics differently in different locations, this is a highly desirable feature for our model. [20,21].

We used a two-dimensional piecewise constant spline function to estimate the spatially varying covariate effect $\beta(\cdot)$ [46]. We assume

$$\beta(\ell_s) = \sum_{i=1}^{m_1} \sum_{j=1}^{m_2} \phi_{ij} B_{i1}(\ell_{s1}) B_{j2}(\ell_{s2})$$

where $B_{11}(\cdot), B_{21}(\cdot) \ldots B_{m_11}(\cdot)$, and $B_{12}(\cdot), B_{22}(\cdot), \ldots, B_{m_22}(\cdot)$ are piecewise constant B-spline basis functions in the north-south and east-west directions respectively, with $k_1$ and $k_2$ regularly spaced interior knots respectively; $m_1 = k_1+1$ is the number of basis functions in the north south direction and $m_2 = k_2+1$ is the number of basis functions in the east-west direction, and $\boldsymbol{\phi} = (\phi_{11}, \phi_{12}, \ldots, \phi_{m_1m_2})\prime$ is the corresponding vector of spline basis coefficients for the spatially varying coefficients [47]. Note that in the case of two dimensional piecewise constant splines, the knots partition the domain into an $m_1 \times m_2$ grid, and for $\boldsymbol{\ell}_s$ in the (i,j)th grid cell, $\beta(\boldsymbol{\ell}_s) = \phi_{ij}$.

Spline models are sensitive to the number of knots used to specify the spline basis functions. Using too few knots can lead to an underfitted model, while using too many knots can result in an overfitted model [46]. To avoid these issues, we specify a large number of knots and use Bayesian two-dimensional penalized splines (P-splines) to deter overfitting. Specifically, we place intrinsic conditional autoregressive (CAR) priors on the spline basis coefficients, which discourage abrupt changes in the spline estimator [48]. Thus, we assume

$$\boldsymbol{\phi}|\sigma^{-2} \sim N(\mathbf{0}, \sigma^{-2}(\boldsymbol{D_\phi} - \boldsymbol{W_\phi}))$$

where $N(\boldsymbol{\mu}, \boldsymbol{\Sigma^{-1}})$ denotes a multivariate normal distribution with mean $\boldsymbol{\mu}$ and *precision* matrix $\Sigma^{-1}$, and $\boldsymbol{W_\phi}$ is an $m_1 m_2 \times m_1 m_2$ binary adjacency matrix with a row and column for each spline basis coefficient. Specifically, $\boldsymbol{W_\phi}(q,q') = 1$ if the coefficients corresponding to row $q$ and column $q'$ are adjacent and 0 otherwise. Two spline coefficients $\phi_{ij}$ and $\phi_{i'j'}$ are adjacent if $i = i'$ and $|j-j'| = 1$ or $|i-i'| = 1$ and $j = j'$, i.e. two coefficients are adjacent (for piecewise constant splines) if the corresponding grid cells share an edge. Finally, $\boldsymbol{D_\phi}$ is a diagonal matrix with $\boldsymbol{D_\phi}(q,q) = \sum_{q'=1}^{m_1 m_2} \boldsymbol{W_\phi}(q, q')$, and $\sigma^{-2}$ is an unknown positive parameter. For more on CAR models, see Besag, 1974 [49] or Banerjee et al., 2015 [44]. Since their initial development by Lang & Brezger (2004) [48], two dimensional Bayesian penalized splines have been widely used in spatial models, both as spatially varying coefficients and as random effects; applications include forestry [50], disease mapping [51], and meteorology [52].

Our species richness data had strong spatial autocorrelation. Such dependence must be accounted for in the model in order for estimation and inference to be reliable. Towards this end, we include spatial random effects (the $\gamma(\cdot)$ terms) in our model. The spatial random effects $\gamma(\cdot)$ are modeled at the HUC2 region with an intrinsic CAR prior. Define the 18-dimensional vector $\boldsymbol{\varphi} = (\varphi_1, \varphi_2, \ldots, \varphi_{18})'$ to have one entry for each HUC2 region. For location $\boldsymbol{\ell_s}$ in HUC2 region i, define $\gamma(\boldsymbol{\ell_s}) = \varphi_i$, so that all locations in the same HUC2 region share the same random effect. An intrinsic CAR prior is used to model the spatial dependence in the random effects. Specifically, we assume

$$\boldsymbol{\psi}|\tau^{-2} \sim N(\mathbf{0}, \tau^{-2}(\boldsymbol{D_\psi} - \boldsymbol{W_\psi}))$$

where $\boldsymbol{W_\psi}$ is an 18×18 binary adjacency matrix with a row and column for each HUC2 region, whose row-column entries are 1 when the corresponding HUC2 regions share a boundary and 0 otherwise. Similarly, $\boldsymbol{D_\psi}$ is a diagonal matrix with $\boldsymbol{D_\psi}(q,q) = \sum_{q'=1}^{18} \boldsymbol{W_\psi}(q, q')$, and $\tau^{-2}$ is an unknown positive parameter. To ensure that the spatial random effects were necessary in the model, we also fit the model to each dataset without including the random effects. The residuals from these models exhibit strong spatial dependence, indicating that the model without random effects fails to adequately account for the spatial dependence in the data. Including the random effects resolves these issues.

To complete the specification of our Bayesian model, the following weakly informative prior distributions are placed on the remaining parameters:

$$\boldsymbol{\alpha} \sim N(\mathbf{0}, \sigma_\alpha^{-2}\boldsymbol{I}), \quad \sigma^{-2} \sim IG(\alpha_\sigma, \beta_\sigma), \quad \tau^{-2} \sim IG(\alpha_\tau, \beta_\tau),$$

where $IG(\alpha, \beta)$ denotes the inverse gamma distribution with shape parameter $\alpha$ and scale parameter $\beta$, $\sigma_\beta^2 = 1000$, and $\alpha_\sigma = \beta_\sigma = \alpha_\tau = \beta_\tau = 1$. Weakly informative priors have little influence on the analysis and allow the posterior distribution to be governed by the observed data.

A metropolis-based Markov chain Monte Carlo (MCMC) routine was used to obtain a sample of the parameters from the posterior distribution using adaptive proposal distributions to improve convergence [53,54]. To improve numerical performance, spline basis coefficients

corresponding to grid cells falling outside of the range of the observed data (i.e. the contermi-nous U.S.) are removed from the model prior to fitting, resulting in a total of Q = 861 spline basis coefficients. The MCMC routine was run for 15,000 iterations, and the first 10,000 were discarded as burn in. Convergence was assessed via trace plots. Depending on the model, fit-ting took 19.7–23.6 hours on Dell Precision 3630 Tower with an Intel® Xeon® E-2186G CPU @3.80 Ghz, 3792 Mhz, with 6 cores and 64 GB of RAM. Point estimates were generated using the posterior mean of the MCMC sample, after discarding the first 10,000 samples as burn in. Significance is assessed with 95% credible intervals calculated from the MCMC out-put. Additional details regarding the full conditional distributions of all parameters, step by step outline of the MCMC routine, and convergence assessment are provided in S1 Appendix. For more on convergence assessment and Bayesian inference, see Gelman et al., 2014 [55].

## Results

### Proportional wetland cover

The proportional wetland cover varied from 0.0 to 5841.0 ha/100 km$^2$. The highest densities of wetlands were concentrated in the southeastern U.S. including portions of Alabama, Georgia, Florida, and North and South Carolina (Fig 1). Florida had the largest percentage of its land area containing high densities of wetlands, especially in far northern Florida and in and around Everglades National Park. The flood plain of the lower Mississippi Valley also con-tained a major concentration of wetlands, along with portions of northern Minnesota and Wisconsin (Fig 1). The western U.S. was less dynamic in spatial heterogeneity of wetland cover with areas of <100 ha/100km$^2$ in the Mojave and Sonoran Desert basins.

### Species richness models

Piecewise constant models for the four taxa groups showed regional hotspots where propor-tional wetland cover was positively, and in some instances, negatively correlated with species richness. Overall, we did not detect consistent continental-scale relationships between wetland cover and species richness but did identify regional correlations at the U.S. EPA level II and III ecoregions (Fig 2). Birds, reptiles and endemic species groups all showed large spatial areas of statistically significant associations while mammals and reptiles showed relatively larger nega-tively significant areas (Fig 3).

Spatial variability of amphibian species richness appeared highly associated to proportional wetland cover in a collection of ecoregions. The Great Plains and the Mojave and Sonoran Desert basins had the highest positive estimates of wetland cover effect (Fig 3). Wetland cover was also significantly and positively associated with amphibian species richness within the Piedmont and the Southeastern Plains of the southeastern U.S. and areas of the Erie Plains and Northern Lakes & Forests ecoregions of Ohio, Michigan, and Wisconsin (Fig 3A). Amphibian richness had negative or no correlation with wetland cover in much of the higher elevations of the Rocky and Appalachian Mountains, the Mississippi Alluvial Plain, and the Cold Deserts regions of Nevada and southern Idaho (Fig 2). Finally, there was a negative correlation with proportional wetland cover for most of Florida including the Southern Florida Coastal Plain ecoregion. All fixed effects covariates were statistically significant, precipitation had the highest relative magnitude of effect (Table 1). Elevation was negatively associated with amphibian spe-cies richness while precipitation and temperature were positively associated with species richness.

Bird species richness had roughly similar correlations with amphibian species richness including in the Mojave and Sonoran Desert Basins, the Great Plains and portions of the Southeastern coastal plains. This included some of the highest positive coefficient values in

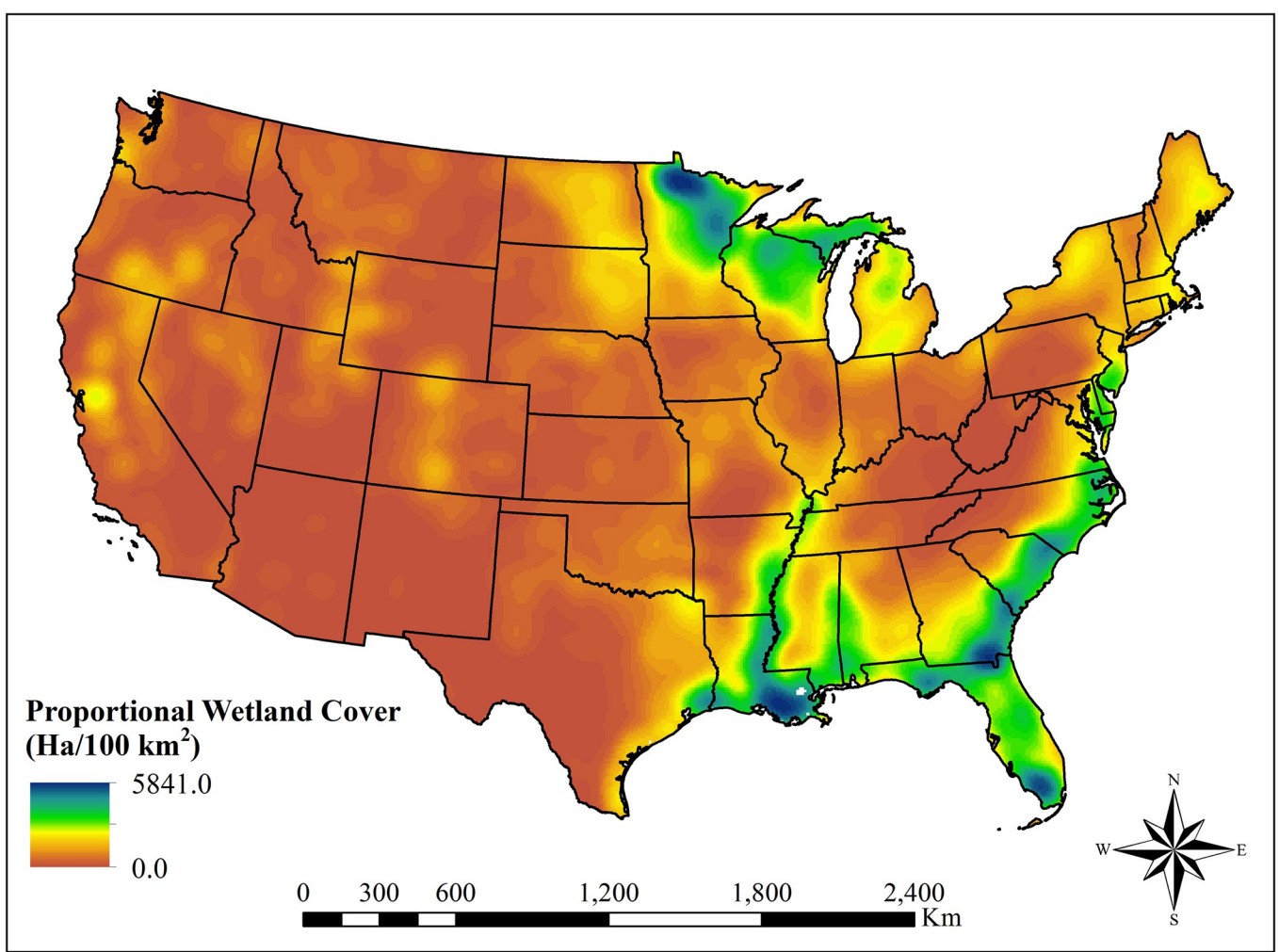

**Fig 1. Proportional wetland coverage in the conterminous United States.** Proportional wetland cover was derived from the U.S. Fish and Wildlife's National Wetland Inventory. The highest wetland densities were modeled in Mississippi Alluvial Plain, the Southern Coastal Plain, and northern Minnesota. Coverage values are hectares/100 km$^2$.

Nebraska and Kansas. In addition, there was little or negative correlation in much of the higher elevations of the Rocky Mountains and the Mississippi Alluvial Plain. However, the correlation with wetland cover was significantly positive in the Appalachian Mountains, Ohio River basin and most of Florida, opposite from amphibians (**Fig 3**). Elevation and temperature were both positively significant fixed effects and precipitation was positive but not significant in the bird species richness model (**Table 1**).

Mammal species richness was not as highly correlated with proportional wetland cover in comparison to other taxa groups. Similar to amphibians and birds, mammalian species richness was positively correlated with wetland cover in the Great Plains and Mojave Desert Basin and negatively correlated in the Rocky Mountains and Mississippi Alluvial Plain. There was also no or negative correlation in much of the southeastern U.S. (**Fig 3**). Mammal species richness was negatively correlated with wetland cover in the Central Appalachians and Blue Ridge ecoregions of the eastern U.S. and the Cordillera of the Pacific Northwest. The elevation fixed effect coefficient was significantly positive and by far the highest estimated magnitude

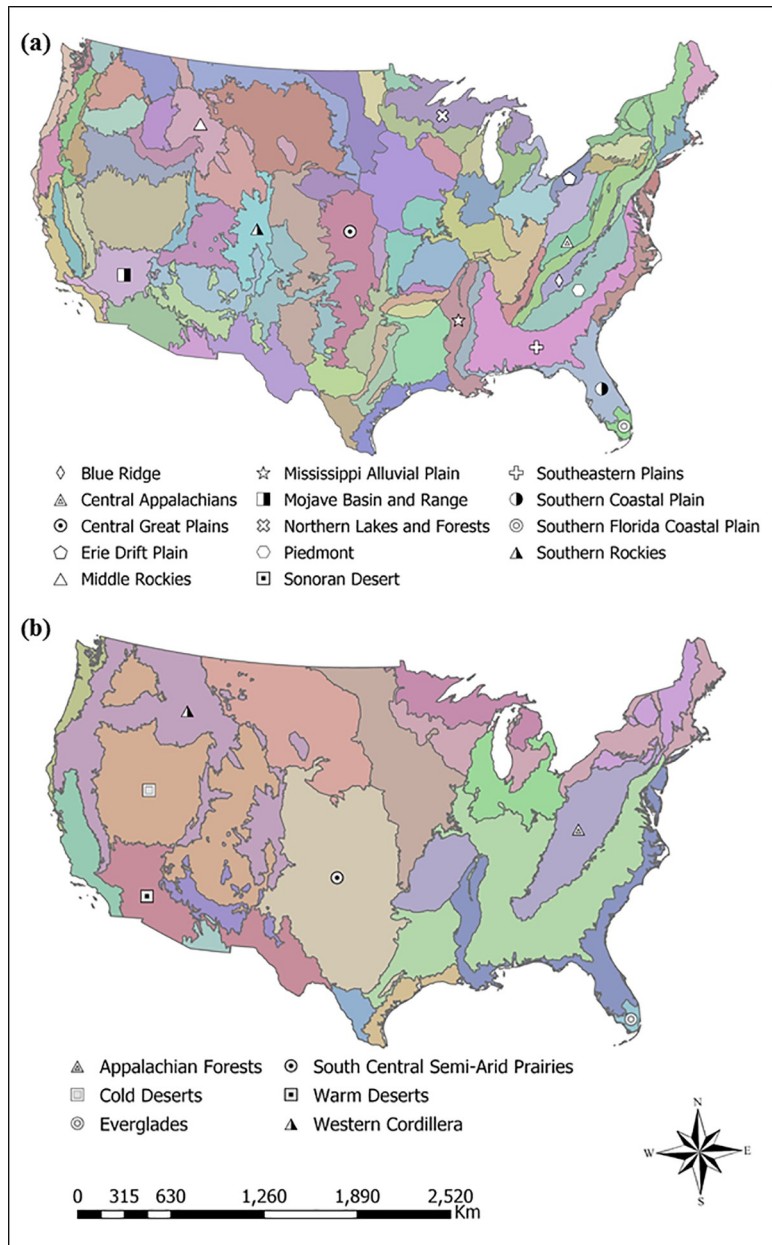

**Fig 2. Level III and II Ecoregions of the United States.** (**a**) Level III and (**b**) level II ecoregions of the United States as defined by the U.S. Environmental Protection Agency. Ecoregions labeled in the figure are referenced in the text.

compared to the other fixed effect covariates (Table 1). Precipitation and temperature were negatively significant in the mammal richness model.

Reptile species richness had the highest positive estimates of the wetland coefficient in the Southern and Middle Rockies ecoregions and the Puget Lowland of Washington (Fig 3). This was in sharp contrast to much of the southwestern U.S. and Central Great Plains which were negatively correlated with reptile species richness. Similar to bird species richness, reptiles were significantly correlated with wetland cover in the Piedmont and Central Appalachian Mountains. All fixed effects covariates were positively significant, the temperature coefficient had the highest magnitude of effect (Table 1).

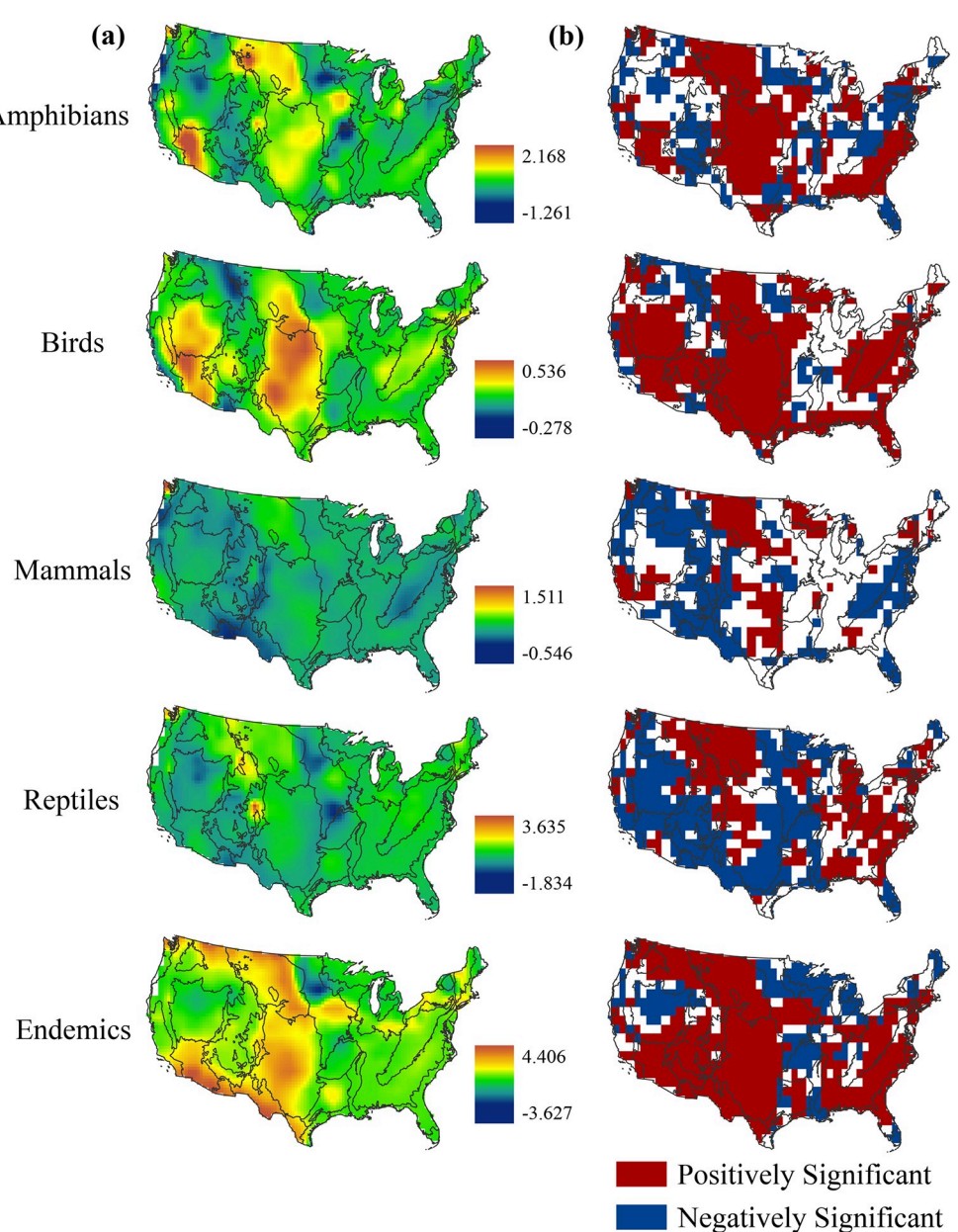

**Fig 3. Spatially varying coefficient estimates of wetland coverage from Bayesian spatial Poisson models.** (**a**) Spatially varying coefficients of wetland cover for the five taxonomic groups varied across the conterminous U.S. (**b**) Spatial variations in species richness and proportional wetland coverage were positively (red) and negatively (blue) significant at the 95% credible level in different ecological regions of the study area. Areas in white are not statistically significant. All maps are presented with the level II ecoregions of the conterminous U.S.

## Endemic species model

The cumulative raster of terrestrial endemic species displayed a major concentration of endemics in the southeastern Atlantic and Gulf of Mexico coasts (**Fig 4**). The number of endemic species quickly decreased to the west of 95˚ longitude and north of 36˚ latitude. Outside of the major concentration of endemic species in the southeastern U.S., the majority of the conterminous U.S. had fewer than 10 endemic species from these four taxa categories.

**Table 1. Bayesian spatial Poisson model coefficient estimates for fixed effect covariates.** Models for the all five taxonomic groups included elevation, precipitation, and temperature as fixed effect covariates. All covariates were significant except for the bird species richness and precipitation.

| Groups & Covariates | Mean | Lower 95% CI | Upper 95% CI |
|---|---|---|---|
| **Amphibians** | | | |
| Elevation | -0.0539 | -0.0632 | -0.0424 |
| Precipitation | 0.1701 | 0.1635 | 0.1763 |
| Temperature | 0.0230 | 0.0213 | 0.0252 |
| **Birds** | | | |
| Elevation | 0.0290 | 0.0269 | 0.0315 |
| Precipitation | 0.0012 | -0.0005 | 0.0028 |
| Temperature | 0.0090 | 0.0084 | 0.0095 |
| **Mammals** | | | |
| Elevation | 0.0524 | 0.0482 | 0.0568 |
| Precipitation | -0.0090 | -0.0121 | -0.0059 |
| Temperature | -0.0028 | -0.0039 | -0.0021 |
| **Reptiles** | | | |
| Elevation | 0.0447 | 0.0359 | 0.0536 |
| Precipitation | 0.0634 | 0.0575 | 0.0693 |
| Temperature | 0.0800 | 0.0779 | 0.0819 |
| **Endemics** | | | |
| Elevation | 0.1821 | 0.1704 | 0.1955 |
| Precipitation | 0.1874 | 0.1797 | 0.1958 |
| Temperature | 0.0794 | 0.0780 | 0.0817 |

Covariate effects are considered significant if 95% credible intervals (CI) does not cross zero.

Endemic species distribution was positively correlated with proportional wetland cover over the majority of the conterminous U.S. The wetland coefficient was significantly positive across much of the western states with the highest coefficient estimates in the South Central Semi-Arid Prairies and Warm Desert level II ecoregions (**Fig 2**). Further, similar to amphibians, birds, and reptiles, endemic species richness across much of the Southeastern Plains was positively correlated with wetland cover (**Fig 3**). The wetland cover coefficient was negatively correlated in Michigan, Minnesota, and Wisconsin which contain a high concentration of wetlands. All fixed effects covariates were positively significant, with the elevation and precipitation covariates having similar effect magnitudes (**Table 1**).

## Wetland change

Wetland coverage in our models decreased by approximately 481,500 ha from NLCD 2001 to NLCD 2011 (**Fig 5A**). The highest wetland change by hectare and percentage was in the northern Great Plains. There were hotspots of percentage wetland reduction in the Mojave Desert Basin and within other parts of the arid west (**Fig 5B**). In addition, there was substantially less hectares of wetlands in the lower Mississippi Alluvial Plain, much of the Southeastern Plains, and Florida. There appeared to be limited increases in wetland coverage in a few locations including southern Oregon and east Texas (**Fig 5B**).

## Discussion and conclusion

We have found that there is large spatial variation in the relationship between the distribution of biodiversity in the form of species richness and the proportional coverage of wetland

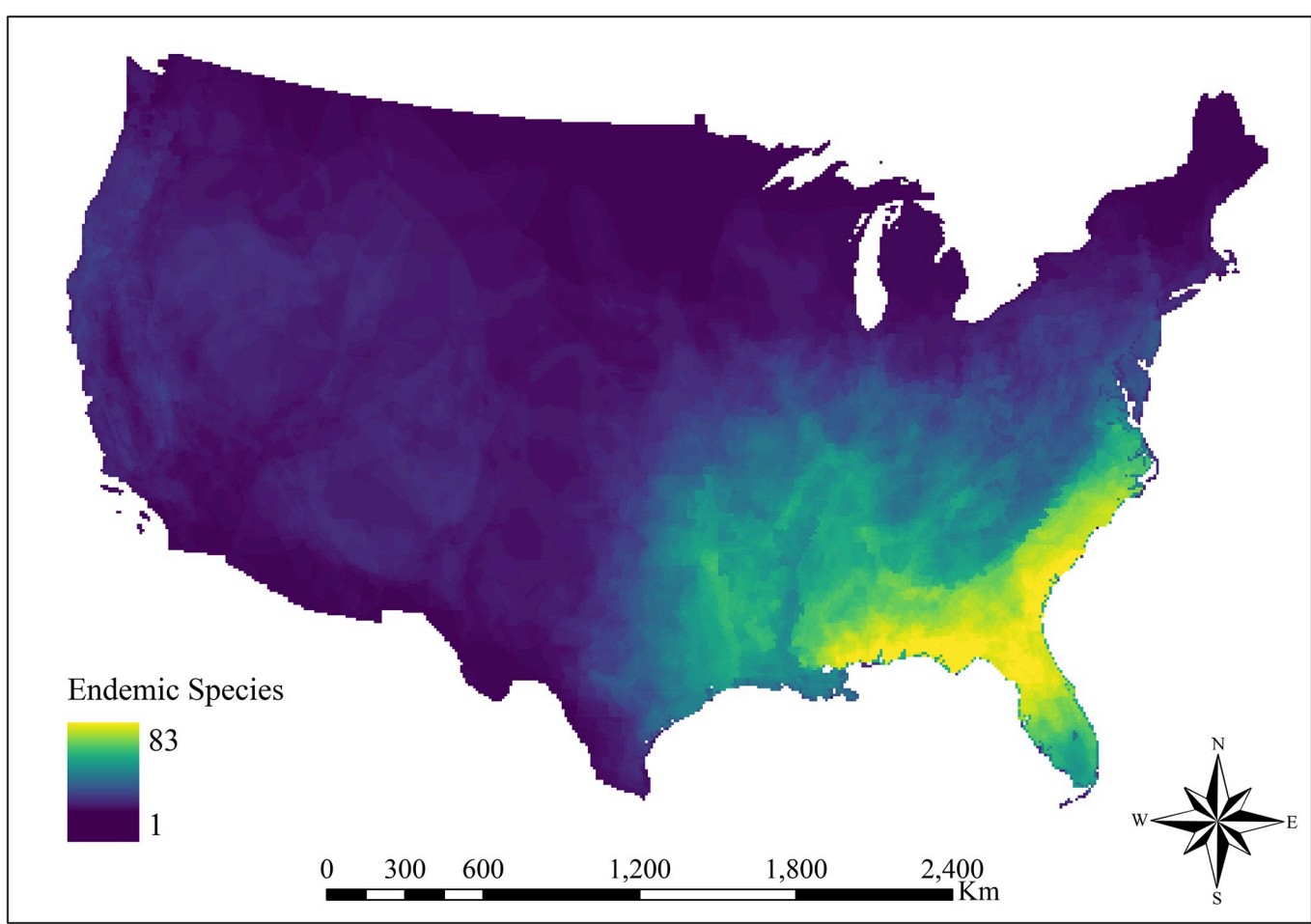

**Fig 4. Cumulative range map of endemic amphibian, bird, mammal, and reptile species in the conterminous U.S.** Most endemic species within the conterminous United States were within the southeastern U.S. The western U.S. had ten or less overlapping endemic species. Endemic species were defined as a species in which the species entire distirbution is within the conterminous U.S.

habitats (**Fig 3**). While we expected this relatively coarse analysis to show a positive continental-scale trend between taxa species richness and proportional wetland cover, we found that the relationship varied more closely to the level II and level III ecoregion scale. Across the five models, certain regions repeatedly stood out as significant hotspots of positive correlation between proportional wetland cover and richness including the Southeastern Plains and Piedmont of the southeast U.S., the Mojave and Sonoran Desert Basins, and the Great Plains. Conversely, the Mississippi Alluvial Plain repeatedly showed no or negative correlation and higher elevation ecoregions including the Southern and Middle Rockies and the Appalachian Forests were negative for amphibians and mammals while positive for birds, reptiles and endemics.

The Southeastern Plains and Piedmont ecoregions contain relatively high species richness and are in a transition area from low wetland cover in the higher elevations of the Appalachian Mountains to high wetland cover in the Atlantic Coastal plain. Identifying these two level III ecoregions as important for wetland conservation is especially paramount as they are at the center of rapid human development that is expected to increase in future decades [56]. Increased human development can lead to the loss and fragmentation of wetland habitat and ultimately the decrease of species richness across taxa [57].

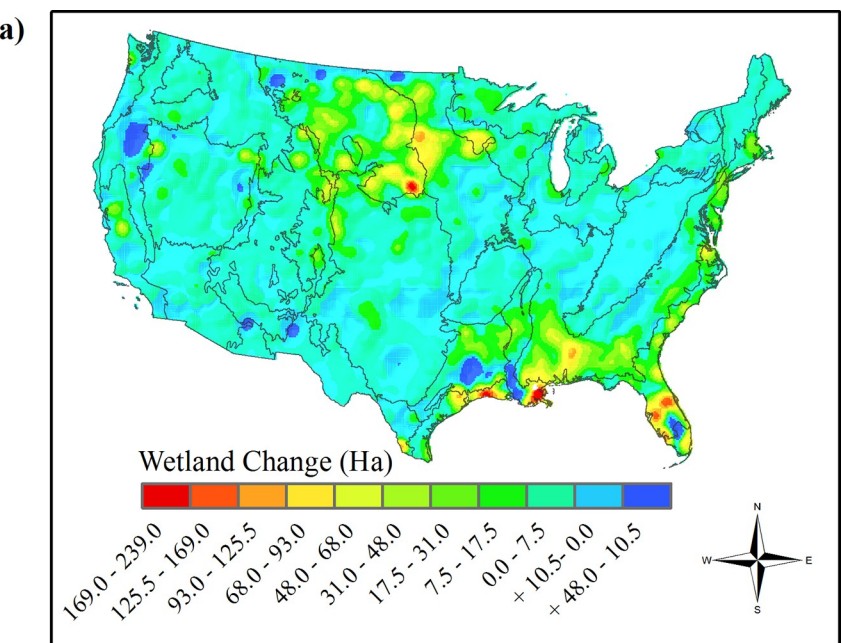

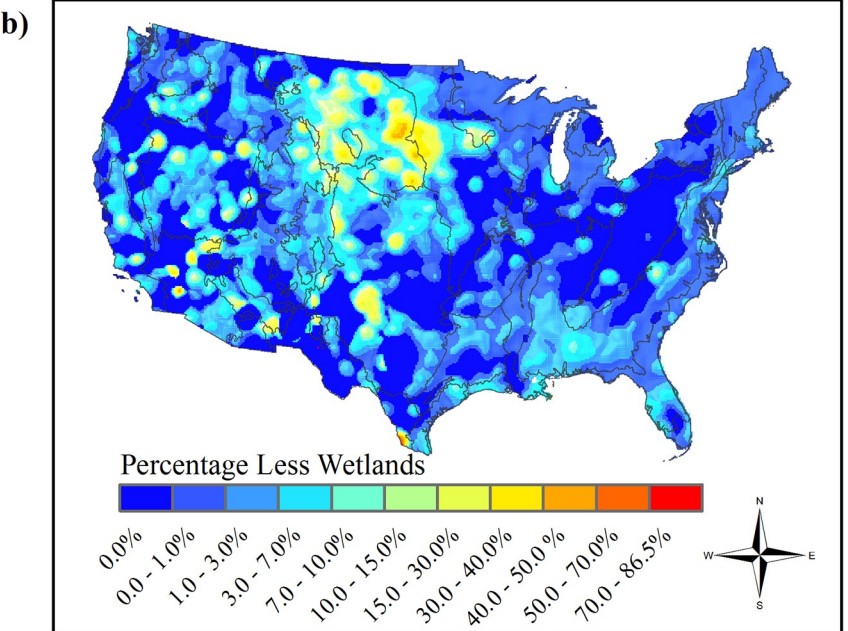

**Fig 5. Wetland change in the conterminous U.S. 2001 to 2011.** Transition per 100 km$^2$ pixel of woody and emergent herbaceous wetlands to other land covers between the National Land Cover Database 2001 and 2011. The highest total hectare change (**a**) occurred in the northern Great Plains, Southeastern and Mississippi Alluvial Plains. Greatest percentage wetland change (**b**) occurred in the northern Great Plains and areas of the desert southwest. There was a reduction of approximately 481,000 ha of wetlands over the ten-year period. Boundary lines are the level II ecoregions of the conterminous U.S.

Though portions of the Appalachian Forests have some of the highest amphibian species richness in the U.S., much of this richness is of Plethodontid salamanders that require wet damp conditions but do not require wetlands for life history function [58–60]. Riparian areas adjacent to first and second-order streams are especially important; however, these small

stream-focused terrestrial habitats are not within the NWI [58,61]. Though wetlands in these mountainous areas are important for the habitat use and conservation of many amphibian species [62,63], overall there is no or a negative relationship between amphibian biodiversity and the proportional cover of wetland habitat in this level II ecoregion. This example of Plethodontid salamanders is important to display the influence that wide ranging life-history strategies at the taxonomic class level have on such a landscape analysis.

The Mojave and Sonoran Desert Basins were positively correlated with all but reptile species richness. Given the aridity of the area and the scarcity of wetlands (e.g., < 30 ha wetlands/ 100 km$^2$), one may assume little correlation between species richness and wetland cover (**Fig 1**). However, species richness values for this ecoregion are additionally near the lowest values in the United States. Research of other deserts have found the highest regional bird and amphibian diversity in isolated wetlands [64,65]. Our results indicate that a small increase in proportional wetland coverage across the desert landscape could be a major determinant of species presence or absence in the exceedingly dry environment. In addition, the Great Plains and the desert southwest had some of the highest positive coefficient values. Conservation and potential restoration of wetlands in these more arid ecoregions could therefore have the highest magnitude of effect on species presence and conservation compared to other areas in the nation.

The Great Plains region is important habitat for resident and migratory birds within the North American Central Flyway [19]. Additionally, restoration of wetland areas within this region is an important determinant in amphibian species richness [66,67]. This is in context to our estimation of wetland change which showed the greatest percentage decrease in wetland area in the Great Plains and in pockets of the desert southwest (**Fig 5**). The loss of these wetlands is likely due to a combination of anthropogenic alteration, climatic variation and other factors, thus some of the wetland change, especially in portions of the Great Plains, is likely temporary [68]. Further comparison with a larger time-series of data will enhance our understanding of wetland change.

Florida contained both the highest levels of proportional wetland cover and some of the highest values of species richness for the five taxa groups. A simple estimation would therefore assume a strong positive correlation; however, our models predicted a negatively significant correlation for amphibians, mammals, reptiles, and endemic species. This counterintuitive finding is likely due to a few interacting factors. Most of the data points were on or adjacent to a point on a data boundary which can influence spatially varying model estimates. In fact, many spatial techniques are known to perform poorly on peninsulas [69,70], and more research is needed for this issue given the rapid increase of landscape-scale spatial biodiversity modeling [71,72]. In addition, species richness values decrease from northern to southern Florida and the data in the region are an extreme outlier making model fit more difficult. The raw data, before considering spatial relationships, suggest that compared to the continental U. S., Florida is a hotspot for both wetland cover and richness across all taxa. But, when considering the peninsular aspect of Florida, boundary effects of spatial analysis, and comparisons across such a large extent, the model estimates should be interpreted cautiously for most of Florida.

Species richness is a basic measure of biological diversity and as a response variable is likely to be less sensitive to environmental change than measures such as community occupancy or abundance [73]. At this coarse of a scale, landscape features that may be vital for rare or declining species may be overwhelmed by the overall landscape pattern that also includes highly abundant species. Therefore, even though we found a negative association in parts of the country between species richness and proportional wetland cover, it does not mean that wetlands in those areas do not serve a vital role in the conservation of wildlife populations. In addition,

there is a temporal mismatch between the ecological forces influencing species distribution and the current distribution of wetlands within the NWI. This mismatch of temporal scales and differences in anthropogenic perturbations that influence these distributions must be recognized for potential impacts on the ultimate inference from our findings. However, the large spatial grain and extent in which we are focusing reduces the influence of this temporal mismatch on our ultimate inference since we capture the general pattern of variation in proportional wetland cover across the sub-continent.

Wetland restoration and maintenance have increasingly become important tenets of U.S. habitat conservation. While this is codified in numerous U.S. legislative acts including the Migratory Bird Conservation Act, Clean Water Act and Wetlands Loan Act and is implemented through several public-private partnerships such as the USFWS Partners for Fish and Wildlife Program [74], there is considerable work needed to increase protections for some wetland classifications such as isolated wetlands. It is a safe assumption that conservation of wetland habitat serves to provide wildlife with temporary or permanent resources, no matter the region. However, focusing on these habitats for biodiversity conservation may not be an equally valuable enterprise across ecological regions. This analysis shows that the spatial correlation between wetland coverage and species richness has tremendous spatial heterogeneity both within and between taxonomic classes. This finding points to the importance for region-based conservation prioritization at a multi-state ecoregional scale, such as the EPA level III ecoregion, rather than those confined only within state boundaries. Political collaboration across states and countries and through public-private partnerships, such as current initiatives within the prairie pothole region of the northern Great Plains (e.g., Prairie Pothole Joint Venture), is required to produce the best spatially-explicit policies to maintain species across the different classes of wildlife [75]. Further analysis of species densities, focal species correlation to wetlands, and species richness in correlation to other wetland metrics will go further in providing direction for regional conservation planners.

## Supporting information

**S1 Appendix. Full conditional distribution and MCMC routine of the Bayesian spatial Poisson models.**
(DOCX)

## Acknowledgments

Support for this study was provided by the Margaret H. Lloyd-SmartState Endowment at Clemson University. We would like to thank Daniel Hanks and Paul Leonard for advice during conceptualization and data analysis. Any use of trade, firm, or product names is for descriptive purposes only and does not imply endorsement by the U.S. Government.

## Author Contributions

**Conceptualization:** Jeremy S. Dertien, Beth E. Ross, Kyle Barrett, Robert F. Baldwin.

**Data curation:** Jeremy S. Dertien, Stella Self.

**Formal analysis:** Jeremy S. Dertien, Stella Self.

**Investigation:** Jeremy S. Dertien.

**Methodology:** Jeremy S. Dertien.

**Project administration:** Jeremy S. Dertien.

**Validation:** Jeremy S. Dertien, Stella Self, Beth E. Ross, Kyle Barrett, Robert F. Baldwin.

**Visualization:** Jeremy S. Dertien, Stella Self.

**Writing – original draft:** Jeremy S. Dertien.

**Writing – review & editing:** Jeremy S. Dertien, Stella Self, Beth E. Ross, Kyle Barrett, Robert F. Baldwin.

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
