## [Decision Letter · Decision Letter 0]

28 Jan 2020

PONE-D-19-34337

The relationship between biodiversity and wetland density varies across regions of the conterminous United States

PLOS ONE

Dear Mr. Dertien,

Thank you for submitting your manuscript to PLOS ONE. After careful consideration, we feel that it has merit but does not fully meet PLOS ONE’s publication criteria as it currently stands. Therefore, we invite you to submit a revised version of the manuscript that addresses the points raised during the review process.

We would appreciate receiving your revised manuscript by Mar 13 2020 11:59PM. To enhance the reproducibility of your results, we recommend that if applicable you deposit your laboratory protocols in protocols.io, where a protocol can be assigned its own identifier (DOI) such that it can be cited independently in the future. For instructions see: http://journals.plos.org/plosone/s/submission-guidelines#loc-laboratory-protocols

We look forward to receiving your revised manuscript.

Kind regards,

Daehyun Kim, Ph.D.

Academic Editor

PLOS ONE

Additional Editor Comments:

Dear Authors,

Thank you for submitting your work to PLOS ONE. I hereby submit my decision as Academic Editor on PONE-D-19-34337 "The relationship between biodiversity and wetland density varies across regions of the conterminous United States." The two reviewers I invited were generally positive about this submission and I concur with them. Especially, Reviewer 1 was very much satisfied with the work and recommended minor revisions. The second reviewer (a spatial statistician) was a bit concerned with the methodological section, but I believe that you will be able to address the issues raised.

Additionally, I would like you to address the following concerns of mine in the revision:

(1) In the last paragraph of the Intro section, there are a set of predictions (or hypotheses) presented. Please provide some background information as to how you came up with these expectations.

(2) Fig. 3 -- (a) and (b) are missing. Also, please provide appropriate legends for each panel.

(3) Fig. 4 -- The legend looks upside down. Please change it so that "1" goes to the bottom and "83" comes to the top.

(4) Figs. 3 and 5 -- The 48 state boundaries are useless in these maps, aren't they? Why don't you present the ecoregion boundaries instead?

Reviewers' comments:

Reviewer's Responses to Questions

**Comments to the Author**

1. Is the manuscript technically sound, and do the data support the conclusions?

Reviewer #1: Yes

Reviewer #2: Yes

2. Has the statistical analysis been performed appropriately and rigorously? 

Reviewer #1: Yes

Reviewer #2: Yes

3. Have the authors made all data underlying the findings in their manuscript fully available?

Reviewer #1: Yes

Reviewer #2: Yes

4. Is the manuscript presented in an intelligible fashion and written in standard English?

Reviewer #1: Yes

Reviewer #2: Yes

5. Review Comments to the Author

Reviewer #1: PONE-D-19-34337

The relationship between biodiversity and wetland density varies across regions of the

conterminous United States

Reviewer comments

I was excited to read this manuscript. The manuscript addresses an interesting topic, the relationship between biodiversity and wetland density across regions of the conterminous United States.

Previous work, such as ‘Status and Trends of Wetlands in the Conterminous United States 2004 to 2009’ (Dahl, 2011) provided baseline information regarding trends in wetland extent and type to facilitate collaborative efforts to assess wetland condition. The geospatial analyses provided insights to regional patterns of wetland loss and indicated that particular regions of the conterminous United States experienced different rates of wetland loss depending on many factors (Dahl, 2011;69). This study digs a bit deeper to address the spatially explicit relationship between wetland density and biodiversity in terms of species distribution of amphibians, birds, mammals, reptiles, and terrestrial endemic species. The authors use National Wetland Inventory to model wetland density for the conterminous United States and the National Land Cover Database to estimate wetland change between 2001 and 2011, and investigate the potential relationship in terms of Bayesian spatial Poisson model. This study builds nicely on past contributions in the field and is a much-anticipated advancement that extends our understanding of the connections between biodiversity and wetland density. In that regard, the manuscript makes a useful contribution to the literature.

Comments:

In the introduction (line 65) it might be useful to give more explanation in relation to the purpose of study.

Line (80) Is it due to?

Line (112) Is it hydrologic unit codes (HUC) or hydrologic units (HUC)? How many HUC-2 regions in the conterminous US? What about the next level unit, HUC-4 to analyze the data?

Line (114) needs a citation or rationale regarding cutoff of 0.7 as an indicator of correlation.

Line (296) Consider replacing National Land Cover Database with NLCD.

Line (301) Consider replacing Discussion with Discussion and Conclusion.

In general, the work of Dahl may be relevant:

Dahl, T.E. 2011. Status and trends of wetlands in the conterminous United States 2004 to 2009.

U.S. Department of the Interior; Fish and Wildlife Service, Washington, D.C. 108 pp.

Dahl, T.E. 2006. Status and trends of wetlands in the conterminous United States 1998 to 2004.

U.S. Department of the Interior; Fish and Wildlife Service, Washington, D.C. 112 pp.

Reviewer #2: I read their manuscript very interestingly. This paper was pretty well written and easy to understand.

However, to publish this manuscript in Plos One, two points should be addressed at the revision.

1. In their statistical analysis section, they state that they use Bayesian spatial Poisson model for describing the effect of wetland density on the species of wildlife. However, I think the detailed explanation of the models are missing and it should be required the justification of Bayesian spatially varying Poisson model, especially, the use of the piecewise constant B-spline basis functions. Why we cannot use simpler model? This explanation should be easy in order to make ecologists who have a limited knowledge in statistics understand.

2. In order to estimate parameter, they use the MCMC sampler. My questions are given followings: (1) What MCMC sampler they use? Generally, implementing the MCMC sampler to the spatial model is not straightforward. Did you really use the simple MH sampler? (2) How many iterations you ran and burned-in? (3) Give some examples of trace plots and autocorrelation plots in your response letter. (4) Provide your computation time. And (5) To show the convergence, try to run the chain multiple times and calculate Gelman-Rubin statistics.

6. PLOS authors have the option to publish the peer review history of their article (what does this mean?). If published, this will include your full peer review and any attached files.

Reviewer #1: No

Reviewer #2: No

---

## [Author Response · Author response to Decision Letter 0]

12 Mar 2020

Please see attached "Response to Reviewers" document.

---

## [Decision Letter · Decision Letter 1]

7 Apr 2020

The relationship between biodiversity and proportional wetland cover varies across regions of the conterminous United States

PONE-D-19-34337R1

Dear Dr. Dertien,

We are pleased to inform you that your manuscript has been judged scientifically suitable for publication and will be formally accepted for publication once it complies with all outstanding technical requirements.

With kind regards,

Daehyun Kim, Ph.D.

Academic Editor

PLOS ONE

Additional Editor Comments (optional):

Reviewers' comments:

Reviewer's Responses to Questions

**Comments to the Author**

1. If the authors have adequately addressed your comments raised in a previous round of review and you feel that this manuscript is now acceptable for publication, you may indicate that here to bypass the “Comments to the Author” section, enter your conflict of interest statement in the “Confidential to Editor” section, and submit your "Accept" recommendation.

Reviewer #1: All comments have been addressed

Reviewer #2: All comments have been addressed

2. Is the manuscript technically sound, and do the data support the conclusions?

Reviewer #1: Yes

Reviewer #2: Yes

3. Has the statistical analysis been performed appropriately and rigorously? 

Reviewer #1: Yes

Reviewer #2: Yes

4. Have the authors made all data underlying the findings in their manuscript fully available?

Reviewer #1: Yes

Reviewer #2: Yes

5. Is the manuscript presented in an intelligible fashion and written in standard English?

Reviewer #1: Yes

Reviewer #2: Yes

6. Review Comments to the Author

Reviewer #1: The authors did a good job by taking into account the reviewers’ comments. I was delighted to see how carefully my comments were included in the revision and the author responses. From my view, the manuscript is now ready for publication.

Reviewer #2: (No Response)

7. PLOS authors have the option to publish the peer review history of their article (what does this mean?). If published, this will include your full peer review and any attached files.

Reviewer #1: No

Reviewer #2: No

---

## [Editor Report · Acceptance letter]

22 Apr 2020

PONE-D-19-34337R1 

The relationship between biodiversity and proportional wetland cover varies across regions of the conterminous United States 

Dear Dr. Dertien:

I am pleased to inform you that your manuscript has been deemed suitable for publication in PLOS ONE. Congratulations! Your manuscript is now with our production department. 

With kind regards,

on behalf of

Dr. Daehyun Kim 

Academic Editor

PLOS ONE